# A roadmap for bootstrapping critical gauge theories: decoupling operators of conformal field theories in $d > 2$ dimensions

**Yin-Chen He[1]⋆, Junchen Rong[2]† and Ning Su[3]‡**

**1** Perimeter Institute for Theoretical Physics, Waterloo, Ontario N2L 2Y5, Canada
**2** DESY Hamburg, Theory Group, Notkestraße 85, D-22607 Hamburg, Germany
**3** Department of Physics, University of Pisa, I-56127 Pisa, Italy

⋆ yinchenhe@perimeterinstitute.ca, † junchenrong@gmail.com, ‡ suning1985@gmail.com

## Abstract

We propose a roadmap for bootstrapping conformal field theories (CFTs) described by gauge theories in dimensions $d > 2$. In particular, we provide a simple and workable answer to the question of how to detect the gauge group in the bootstrap calculation. Our recipe is based on the notion of *decoupling operator*, which has a simple (gauge) group theoretical origin, and is reminiscent of the null operator of $2d$ Wess-Zumino-Witten CFTs in higher dimensions. Using the decoupling operator we can efficiently detect the rank (i.e. color number) of gauge groups, e.g., by imposing gap conditions in the CFT spectrum. We also discuss the physics of the equation of motion, which has interesting consequences in the CFT spectrum as well. As an application of our recipes, we study a prototypical critical gauge theory, namely the scalar QED which has a $U(1)$ gauge field interacting with critical bosons. We show that the scalar QED can be solved by conformal bootstrap, namely we have obtained its kinks and islands in both $d = 3$ and $d = 2 + \epsilon$ dimensions.



# 1  Introduction

Coupling gapless particles with gauge fields is one of the few known ways to obtain an interacting conformal field theory in dimensions $d > 2$. These gauge theory type of CFTs have interesting applications in both the high energy [1–3] and condensed matter physics. In condensed matter system, such CFTs describe phase transitions or gapless phases beyond conventional Landau's symmetry breaking paradigm [4–12], and they have interesting properties such as fractionalization and long-range entanglement. Understanding such CFTs may pave the way towards several long-standing problems in condensed matter, including critical quantum spin liquids [6–8] and plateau transitions of fractional quantum Hall states [9–12].

Compared to the Wilson-Fisher (WF) CFTs, these gauge theory CFTs are poorly understood. Recently, conformal bootstrap [13] became a powerful technique to study CFTs in dimensions higher than $2d$ [14–24] (see a review [25]). The numerical bootstrap obtained critical exponents of $3d$ Ising [14] and $O(2)$ WF [23] with the world record precision, and importantly, has solved the long-standing inconsistency between experiments and Monte-Carlo simulations of $O(2)$ WF [23] as well as the cubic instability of $O(3)$ WF [24]. However, so far the gauge theory CFTs resist to be tackled by bootstrap [26–29]. The main challenge is built in the fundamental philosophy of bootstrap, namely characterizing a theory without relying on a specific Lagrangian. More concretely, in a bootstrap study one typically inputs the global symmetry of the theory, and utilizes the consistency of crossing equations to constrain or to compute the scaling dimensions of operators in certain representations of the global symmetry. For a Wilson-Fisher type of CFT, it is believed that one can uniquely define it by specifying its global symmetry as well as the representation of the order parameter, i.e., the lowest lying operators. In contrast, gauge theories with distinct gauge groups could have similar or even identical global symmetries. Their lowest lying operators would sit in the same representation and have similar scaling dimensions. Therefore, it is unclear how to detect the gauge group in a typical bootstrap calculation.

As a concrete example, one can consider a family of theories described by $N_f$ flavors of two-component Dirac fermions coupled to a $U(N_c)$ gauge field in $d = 3$ dimensions. For a given color number $N_c$, most theories in the infrared (IR) will flow into CFTs when $N_f$ is larger than a critical value $N_f^*$. In other words, for a large enough $N_f$ there will be a number of distinct CFTs that correspond to different $N_c$'s. These CFTs have identical global symmetries, i.e. $(SU(N_f) \times U(1)_{top})/\mathbf{Z}_{N_f}$.[1] The $SU(N_f)$ corresponds to the flavor symmetries of Dirac fermions, while the $U(1)_{top}$ symmetry corresponds to the $U(1)$ gauge flux conservation of $U(N_c)$ gauge group. The most important low lying (scalar) operators are 1) fermion bilinears which are $SU(N_f)$ adjoint but neutral under $U(1)_{top}$, its scaling dimension is $\Delta = 2 + O(1/N_f)$; 2) $2\pi$ monopole operators which are charged under $U(1)_{top}$ and also carries a non-trivial representation of $SU(N_f)$ (which is independent of $N_c$), its scaling dimension is $\Delta = 0.265 N_f - 0.0383 - 0.516(N_c - 1) + O(1/N_f)$ [30]. There were efforts to bootstrap

---

[1]For the precise global symmetry one may need to further quotient out certain discrete symmetries. Such global part of the global symmetry is not important for our discussion.

the 4pt of either fermion bilinears [26, 28] or monopole operators [27], but no unambiguous signature of gauge theories is found.

In $d = 2$ dimensions it is pretty common that distinct CFTs have the same global symmetry. However, numerical bootstrap successfully detects some of these CFTs, including the $2d$ Ising CFT out of minimal models [31] and the $SU(2)_1$ Wess-Zumino-Witten (WZW) CFT out of $SU(2)_k$ WZW theories [32, 33]. It is found that the Ising CFT ($SU(2)_1$ WZW) sits at the kink of numerical bounds, while its cousins in the minimal model ($SU(2)_k$ WZW) saturate the numerical bounds on the right (left) hand side of the kink. More interestingly, the phenomenon that these CFTs appear at kinks of bootstrap bound is closely related to the existence of a family of CFTs sharing the same global symmetry and similar operator spectrum. Compared to their cousins, the Ising CFT and $SU(2)_1$ WZW are special because they have null operators at low levels. These null operators will lead to some non-analyticity in the numerical bound, resulting in a kink [15, 33, 34].

The examples in $2d$ suggest that, the existence a family of cousins with the same global symmetries is not an obstacle of bootstrapping a CFT, it could instead guide us to find the right condition (i.e. null operator condition) to bootstrap the CFT of interest. Theoretically, the existence of the null operator at a certain level can serve as a defining feature of 2d minimal model [35]. One cannot help to wonder if a similar physics also exists for higher dimensional CFTs, and can it be further utilized in the bootstrap study? We provide a positive answer to this by exploring gauge theories and in particular, their relations with the $2d$ WZW CFTs. We will show that in gauge theories there exists a family of operators, we dub decoupling operators, that are reminiscent of null operators of $2d$ WZW CFTs in higher dimensions. Similar to the Kac-Moody algebra, the structure of decoupling operators of gauge theories are sensitive to the representations of global symmetry. Moreover, the color number $N_c$ of the gauge group plays the role of the WZW level (i.e. $k$) in WZW CFTs.

We also explore another related observation in higher dimensional CFTs, namely the equation of motion (EOM) can lead to the phenomenon of operator missing in the CFT spectrum [15, 16, 36, 37]. Theoretically, it was understood that as the consequence of the EOM of $\phi^4$ theory, i.e. $\Box\phi = g\phi^3$, $\phi^3$ becomes a descendent of $\phi$. In other words, the operator $\phi^3$ becomes missing in the primary operator spectrum of WF CFTs. This structure can further serve as an algebraic definition of WF CFTs in $4-\epsilon$ dimensions [36]. Numerically, one can also impose the condition of $\phi^3$ being missing by adding a large gap above $\phi$ in the $O(N)$ vector channel, this is indeed how one obtains the famous bootstrap island of WF CFTs [14, 16]. We will push this idea further by exploring the consequence of EOMs on high level missing operators. Such higher level missing operators are actually rather straightforward to visualize. For example, it is natural to expect that $\phi(\Box\phi - g\phi^3)$ is missing as well. We will elaborate more on this and its consequence in the main text.

To be concrete, we will discuss the idea of decoupling operators and their bootstrap application in the context of a prototypical gauge theory, namely the scalar QED. It is described by $N_f$ flavor critical bosons coupled to a $U(1)$ gauge field,

$$\mathcal{L} = \sum_{i=1}^{N_f} |(\partial_\mu - iA_\mu)\phi_i|^2 + m^2|\phi|^2 + \frac{g}{4}|\phi|^4 + \frac{1}{4e^2}F_{\mu\nu}^2. \tag{1}$$

The global symmetry of the scalar QED is $PSU(N_f) = \frac{SU(N_f)}{Z_{N_f}}$. One fundamental (gauge invariant) operators of this theory are the boson bilinear $\bar{\phi}^i\phi_j - \delta^i_j/N\bar{\phi}^k\phi_k$ and $\bar{\phi}^i\phi_i$, which are in the $SU(N_f)$ adjoint and singlet representation, respectively. This theory is dual to the $CP^{N_f-1}$ model, i.e., a non-linear sigma model (NLσM) on the target space $CP^{N_f-1} \cong \frac{U(N_f)}{U(N_f-1)\times U(1)}$. Within the NLσM formulation, one can access the scalar QED fixed point using the $2+\epsilon$

expansion [38–40]. It is worth noting that in $d = 3$ dimensions, there is one extra emergent symmetry called $U(1)_{top}$, which corresponds to the flux conservation of the gauge field. There will be a new type of primary operators, called monopoles [41], that are charged under $U(1)_{top}$. In this paper, we will not study monopole operators.

For a large enough $N_f$, the scalar QED in $2 < d < 4$ dimensions will flow into an interacting CFT as one tunes the mass of bosons to a critical value. In a given dimension $d$, there exists a critical $N_f^*(d)$ below which the scalar QED fixed point will disappear by colliding with the tri-critical QED fixed point (see definition below) [42–45]. In other words, only if $N_f > N_f^*(d)$ the scalar QED will be a real CFT.[2] It is believed that $N_f^*(d)$ monotonically increases with $d$, but its precise form is unknown. From $2 + \epsilon$ and $4 - \epsilon$ expansion, it is found that $N_f^*(d \to 2) \to 0$ [38–40] and $N_f^*(d \to 4) \approx 183$ [42]. It remains an open problem regarding the value of $N_f^*$ in $d = 3$ dimensions [48–50]. The $N_f = 2$ scalar QED in $d = 3$ dimensions is one of the dual descriptions of the widely studied deconfined phase transition in condensed matter literature [4, 5]. There are extensive studies to discuss whether it is truly a CFT in the deep infrared [43, 51–54].

The paper is organized as follows. In Sec. 2 we will introduce the notation of decoupling operators. In particular, In Sec. 2.1 we will show that the null operators of 2$d$ WZW CFTs can be interpreted as decoupling operators of gauge theories. We then discuss decoupling operators of bosonic gauge theories in Sec. 2.2. In Sec. 3 we discuss the consequence of EOMs on the CFT spectrum. In Sec. 4 we will present our numerical results of the scalar QED. In particular, by imposing the information of decoupling operators we show the scalar QED in 3 dimensions (Sec. 4.2) and $2 + \epsilon$ dimensions (Sec. 4.3) can be solved using conformal bootstrap: we have obtained kinks and islands of scalar QED. We will conclude in Sec. 5, and will provide more numerical data in the appendix.

*Note added:* Upon the completion of this work we became aware of an independent work [55] that overlaps with ours.

## 2 Decoupling operators in gauge theories

In this section, we will define what we mean by the decoupling operator and discuss several concrete examples in 2$d$ CFTs and higher dimensional gauge theories.

The decoupling operator of a CFT of interest $\mathcal{A}$ can be defined by embedding $\mathcal{A}$ into a family of CFTs that share the same global symmetry and similar operator spectrum. Then one can construct a possible continuous interpolation between these different theories, and define decoupling operators as operators that decouple from the theories' spectrum as one continuously tunes to the CFT $\mathcal{A}$. A textbook example is the 2$d$ minimal model $\mathcal{M}_{q,q-1}$ for which one can promote the integer valued $q$ to be real valued, which then interpolates all the minimal models $\mathcal{M}_{q,q-1}$. This is more than a conceptual interpolation, indeed we can explicitly write down a number of crossing symmetric correlation functions that continuously depend on $q$.[3] As one continuously tunes $q$, there are operators decoupled from the spectrum at integer valued $q$. These decoupling operators are indeed null operators for a specific theory $\mathcal{M}_{q,q-1}$ [34]. Similarly, for the 2$d$ WZW CFTs, one can promote the integer valued WZW level $k$ to be real valued, and ask how are operators decoupled as one continuously varies $k$ (see Sec. 2.1 for more details). Different from the example of the 2$d$ minimal model, the decoupling (null) operators are lying in representations that strongly depend on $k$'s: for the $SU(N)_{k=l}$ WZW CFT, all the Kac-Moody primaries in the rank-$m$ symmetric tensor representation with

---

[2]We shall note it is an exception for the $N_f = 1$ scalar QED in 3$d$ as it is dual to the $O(2)$ WF [46, 47].

[3]In the end, a fully consistent solution of all crossing symmetric correlation functions would only admit discrete (integer valued) $q$.

$m > l$ are becoming null operators [56].

Although null operators of 2D CFTs can be defined as decoupling operators, null operators certainly have deeper implications in the algebra of CFTs, e.g. they can act as differential operators that annihilate correlation functions of primary operators. The decoupling operators, on the other hand, may or may not have such fundamental applications in the operator algebra of higher dimensional CFTs. It will be interesting to understand the similarity and difference between 2d null operators and higher dimensional decoupling operators in the future.

## 2.1 Null operator as a decoupling operator: the $SU(N)_k$ WZW CFT

In this section, we will elaborate more on how to view the null operator of 2d CFTs as a decoupling operator in the context of the $SU(N)_k$ WZW CFT. Let us start with a simple case, i.e. $SU(2)_k$ WZW theory. It has a global symmetry $SO(4) \cong SU(2)_L \times SU(2)_R/Z_2$, and its Kac-Moody primary operators $|j, j\rangle$ are in the $SO(4)$ representations $(SU(2)_L, SU(2)_R) = (j, j)$ with $j = 0, 1/2, 1, \cdots, k/2$.[4] So $|1, 1\rangle$ is a Kac-Moody primary of $SU(2)_{k\geq 2}$ WZW CFT, while it becomes null in the $SU(2)_1$ WZW CFT.

Now we create an interpolation between all the $SU(2)_k$ WZW CFTs by promoting the integer valued $k$ to be real valued. More precisely, the four-point correlation function (4pt) of any primary operator of the $SU(2)_k$ WZW CFTs is an analytical function of $k$, so there is no obstacle to promote $k$ to be real valued. For our purpose it is enough to consider the 4pt of the Kac-Moody primary $|1/2, 1/2\rangle$, which is a $SO(4)$ vector and we will call it $\phi_i$:

$$
\langle \phi_i(x_1)\phi_j(x_2)\phi_k(x_3)\phi_l(x_4) \rangle = \frac{1}{x_{12}^{2\Delta_\phi} x_{34}^{2\Delta_\phi}} \left[ \frac{\delta_{ij}\delta_{kl}}{N} G^S[z,\bar{z}] \right.
$$
$$
+ (\frac{1}{2}\delta_{il}\delta_{jk} + \frac{1}{2}\delta_{ik}\delta_{jl} - \frac{1}{N}\delta_{ij}\delta_{kl})G^T[z,\bar{z}]
$$
$$
\left. + (\frac{1}{2}\delta_{il}\delta_{jk} - \frac{1}{2}\delta_{ik}\delta_{jl})G^A[z,\bar{z}] \right]. \tag{2}
$$

Here $N = 4$ and $G^S[z,\bar{z}]$, $G^T[z,\bar{z}]$, $G^A[z,\bar{z}]$ corresponds to the 4pt's in the channels of the $SO(4)$ singlet, rank-2 symmetric traceless tensor, and rank-2 anti-symmetry tensor. The precise form of these 4pt's can be found in textbooks such as [56]. We are primarily concerned with the Kac-Moody primary $|1, 1\rangle$, which is in the channel of rank-2 symmetric traceless tensor. The 4pt corresponding in this channel is,

$$
\frac{G^T[z,\bar{z}]}{((1-z)(1-\bar{z}))^{\frac{1}{2k+4}}} = \frac{2z\bar{z}}{k^2}A[z,\bar{z}] + 2(z\bar{z})^{\frac{2}{k+2}}\left( \frac{\Gamma^2(\frac{1}{k+2})\Gamma^2(\frac{3}{k+2})}{\Gamma^4(\frac{2}{k+2})} - \frac{4\Gamma^2(\frac{-2}{k+2})\Gamma^2(\frac{3}{k+2})}{\Gamma^2(\frac{2}{k+2})\Gamma^2(\frac{-1}{k+2})} \right)B[z,\bar{z}], \tag{3}
$$

with

$$
A[z,\bar{z}] = {}_2F_1(\frac{k+1}{k+2}, \frac{k+3}{k+2}, \frac{2k+2}{k+2}, z)\,{}_2F_1(\frac{k+1}{k+2}, \frac{k+3}{k+2}, \frac{2k+2}{k+2}, \bar{z}),
$$
$$
B[z,\bar{z}] = {}_2F_1(\frac{1}{k+2}, \frac{3}{k+2}, \frac{2}{k+2}, z)\,{}_2F_1(\frac{1}{k+2}, \frac{3}{k+2}, \frac{2}{k+2}, \bar{z}). \tag{4}
$$

Decomposing this 4pt into the global conformal blocks, one obtains the low lying spectrum to be $\Delta = \frac{4}{k+2}, 2, \cdots$. The first operator (denoted as $t$) is nothing but the Kac-Moody primary $|1, 1\rangle$, while the second operator is a global primary obtained by applying Kac-Moody current

---

[4]In this notation, $(1/2, 1/2)$ corresponds to the $SO(4)$ vector, $(1, 1)$ corresponds to the rank-2 symmetric traceless tensor, $(1, 0) \oplus (0, 1)$ corresponds to the rank-2 anti-symmetric tensor.

operator to the vacuum, i.e. $J_L J_R |0, 0\rangle$. We can also work out the OPE square $\lambda^2_{\phi\phi t}$ of $|1, 1\rangle$,

$$\lambda^2_{\phi\phi t} = \frac{\Gamma^2(\frac{1}{k+2})\Gamma^2(\frac{3}{k+2})}{2\Gamma^4(\frac{2}{k+2})} - \frac{2\Gamma^2(\frac{-2}{k+2})\Gamma^2(\frac{3}{k+2})}{\Gamma^2(\frac{2}{k+2})\Gamma^2(\frac{-1}{k+2})}. \tag{5}$$

The above formula is positive definite for $k > 1$, and it vanishes precisely at $k = 1$. In other words the Kac-Moody primary $|1, 1\rangle$ gets decoupled from operator spectrum at $k = 1$. Therefore, in this natural interpolation of $SU(2)_k$ WZW CFTs we can view the null operator $|1, 1\rangle$ of $SU(2)_1$ WZW as a decoupling operator.

The above discussion can be easily generalized to the $SU(N)_k$ WZW CFTs. Interestingly, in the large-$N$ limit we can directly relate the Kac-Moody null operator to the decoupling operator of gauge theories, without relying on any precise knowledge of the correlation function or operator spectrum of the WZW CFTs. The key is to recognize a dual description for the $SU(N)_k$ WZW CFTs, namely a gauge theory with $N$ flavors of 2-component Dirac fermions interacting with a $U(k)$ gauge field. For the case of $k = 1$, this duality can be proved exactly as the $U(1)$ gauge theory is integrable [57]. For a general level-$k$ WZW CFT, there are reasonable evidences suggesting that they are dual to a QCD$_2$ theory (e.g. see [58] and references therein), although the QCD$_2$ is not integrable anymore.

The global symmetry of both the $SU(N)_k$ WZW and the $U(k)$ gauge theory is $SU(N)_L \times SU(N)_R$, so we can consider the (global) primary operator spectrum of these theories in various representations of $SU(N)_L \times SU(N)_R$. Let us warm up with the lowest weight Kac-Moody primary (except for the vacuum), i.e., a bi-fundamental of $SU(N)_L$ and $SU(N)_R$. This operator exists for arbitrary $k$, and its scaling dimension is $\Delta = \frac{N^2-1}{N(N+k)}$, which is $\Delta \approx 1 + O(1/N)$ in the limit of $N \gg k$. In the gauge theory, this operator is nothing but 2-fermion operators, schematically written as $\bar{\psi}^c_l \psi_{r,c}$. We use a convention that the right (left) moving fermion $\psi$ ($\bar{\psi}$) is the fundamental (anti-fundamental) of $U(k)$ gauge field, and $c$ is the index of its $SU(k)$ subgroup. The index $l$ and $r$ refer to the index of $SU(N)_L$ and $SU(N)_R$. So this 2-fermion operator $\bar{\psi}^c_l \psi_{r,c}$ is the $SU(N)$ bi-fundamental and its scaling dimension is $\Delta = 1 + O(1/N)$ in the $N \gg k$ limit. We have matched the lowest primary operators of $SU(N)_k$ WZW with the 2-fermion operators of $U(k)$ gauge theories.

Let us now move to the 4-fermion operators (that are Lorentz scalar) of gauge theories. Such operator can be schematically written as $\bar{\psi}^{c_1}_{l_1} \bar{\psi}^{c_2}_{l_2} \psi_{r_3,c_3} \psi_{r_4,c_4}$. The two left (right) moving fermions shall be totally antisymmetric, so we shall have either the flavor indices or the color indices anti-symmetric, and meanwhile keep the other indices symmetric. We need to further contract the color indices of left and right moving fermions to get a gauge invariant operator. For $k = 1$, however, anti-symmetrizing color indices is not an option, leaving the only possibility to be anti-symmetrizing the flavor indices. Therefore, for $k > 1$ there are two different 4-fermion operators (that are Lorentz scalar) which are in the $SU(N)_L \times SU(N)_R$ representations $A_L A_R$ and $T_L T_R$.[5] What are these operators in the $SU(N)_k$ WZW CFTs? There are nothing but the Kac-Moody primaries in the $A_L A_R$ and $T_L T_R$ channel, whose scaling dimensions are $\Delta = \frac{2(N-2)(N+1)}{N(N+k)}$ and $\Delta = \frac{2(N-1)(N+2)}{N(N+k)}$. In the limit of $N \gg k$, these two scaling dimensions are $\Delta = 2 + O(1/N)$ matching what we expect for 4-fermion operators. On the other hand, when $k = 1$ there is only one 4-fermions operator in the channel $A_L A_R$, as the other operator in the channel $T_L T_R$ becomes a decoupling operator due to the low rank of the gauge group. This again nicely matches the physics of $SU(N)_k$ WZW CFTs, namely at $k = 1$ the Kac-Moody primary in the $T_L T_R$ channel becomes null (the Kac-Moody primary in the $A_L A_R$ channel is still intact). It is straightforward to generalize to other Kac-Moody null operators for higher $k$'s, as well as to other WZW CFTs.

---

[5]Here $T_L T_R$ ($A_L A_R$) refers to rank-2 symmetric (anti-symmetric) tensor of $SU(N)_L$ and $SU(N)_R$.

Therefore, on the phenomenological level null operators of 2d WZW CFT can be understood as decoupling operators in the context of 2d gauge theories. From the gauge theory side, we can also generalize the analysis to higher dimensions. A complexity is that fermion is in the spinor representation of $SO(d)$ Lorentz rotation, which has a strong dependence on the spacetime dimension $d$. It turns out that it is easiest to discuss the idea of decoupling operators in the context of bosonic gauge theories, namely critical bosons coupled to gauge fields. We will discuss it in the following subsection. It is also worth mentioning that, in higher dimensions (e.g. $3d$) one can also make a straightforward connection between fermionic gauge theories and WZW CFTs [59,60]. The details are a bit off the theme of the current paper, we will elaborate more in the Appendix.

## 2.2 Decoupling operator of bosonic gauge theories

In this subsection, we will discuss the decoupling operators of bosonic gauge theories, namely critical bosons coupled to gauge fields. We will explain the idea in the context of $U(N_c)$ gauge theories, and the generalization to other gauge groups $SU(N_c)$, $SO(N_c)$, and $USp(2N_c)$ is rather straightforward.

We can simply start by classifying gauge invariant operators (constructed by bosonic field) in these gauge theories. We denote boson operators as $\phi_{f,c}$ and $\bar{\phi}^{f,c}$, which are $SU(N_f)$ ($U(N_c)$) fundamental and anti-fundamental, respectively. $f = 1, \cdots, N_f$ and $c = 1, \cdots, N_c$ correspond to the flavor and color index. To keep the $U(1) \subset U(N_c)$ gauge invariance, we shall only consider operators like $\bar{\phi}^{f_1,c_1} \cdots \bar{\phi}^{f_m,c_m} \phi_{f_{m+1},c_{m+1}} \cdots \phi_{f_{2m},c_{2m}}$. Among these operators, one should further choose $SU(N_c)$ gauge invariant ones. Let us start with $m = 1$, i.e. boson bilinears $\bar{\phi}^{f_1,c_1} \phi_{f_2,c_2}$. Apparently, to keep $SU(N_c)$ invariance there are only two operators, $\bar{\phi}^{f_1,c_1} \phi_{f_1,c_1}$ and $\bar{\phi}^{f_1,c_1} \phi_{f_2,c_1} - \delta^{f_1}_{f_2}/N_f \bar{\phi}^{f,c_1} \phi_{f,c_1}$, which are the $SU(N_f)$ singlet and adjoint, respectively. Their large$-N_f$ scaling dimensions are $\Delta = 2 + O(1/N_f)$ and $\Delta = d-2 + O(1/N_f)$ for $N_c \ll N_f$.

Things become interesting as one moves to $m = 2$. Let us ask what is the lowest operator in the representation $A^{[f_1,f_2]}_{[f_3,f_4]}$, where both the upper and lower indices are antisymmetric. To construct an operator in this representation, one needs at least 4 bosons, $\bar{\phi}^{f_1,c_1} \bar{\phi}^{f_2,c_2} \phi_{f_3,c_3} \phi_{f_4,c_4}$. If $N_c \geq 2$, one can simultaneously antisymmetrize the flavor indices (i.e. $[f_1,f_2]$, $[f_3,f_4]$) and the color indices (i.e. $[c_1,c_2]$, $[c_3,c_4]$) of $\bar{\phi}^{f_1,c_1} \bar{\phi}^{f_2,c_2}$ and $\phi_{f_3,c_3} \phi_{f_4,c_4}$, and then contract their color indices to get a $SU(N_c)$ gauge invariant operator. This will then give an operator in the required representation, with a scaling dimension $\Delta = 2(d-2) + O(1/N_f)$. When $N_c = 1$, in contrast, antisymmetrizing the color indices of two identical bosons will vanish. So the lowest operator in the required representation shall involve two covariant derivatives, schematically written as $(\bar{\phi}^{f_1} D_\mu \bar{\phi}^{f_2})(\phi_{f_3} D_\mu \phi_{f_4})$. Its scaling dimension is $\Delta = 2(d-2) + 2 + O(1/N_f)$. Therefore, in the $A^{[f_1,f_2]}_{[f_3,f_4]}$ channel the QCD gauge theories ($N_c > 1$) have the spectrum $\Delta = 2(d-2) + O(1/N_f)$, $2(d-2) + 2 + O(1/N_f)$, $\cdots$, while for $N_c = 1$ (e.g. scalar QED) the spectrum is $\Delta = 2(d-2) + 2 + O(1/N_f), \cdots$. In other words,

- *In the $SU(N_f)$ $A^{[f_1,f_2]}_{[f_3,f_4]}$ channel, the lowest operator of $U(N_c > 1)$ gauge theories is decoupling at $N_c = 1$.*

One can easily generalize above discussions to arbitrary $N_c$,

- *In the interpolation between $U(N_c)$ gauge theories, the lowest lying operator in the $SU(N_f)$ anti-symmetric representation $A^{[i_1,\cdots,i_m]}_{[j_1,\cdots,j_m]}$ of $N_c > m-1$ is decoupling at $N_c \leq m-1$.*

This structure of decoupling operators is almost identical to the null operator structure of $2d$ WZW CFTs, and the color number $N_c$ plays the role of WZW level $k$. Similar structures can

also be found in theories with other gauge groups.[6]

# 3  Consequence of the equation of motion

The notion of decoupling operator was formulated by identifying a family of CFTs with the identical global symmetry. In the numerical bootstrap, one can impose gap conditions based on the structure of the decoupling operator to isolate the theory of interest from their cousins. On the practical side, depending on the scheme of bootstrap, one may also need to consider other theories that are consistent with crossing equations being bootstrapped. For example, we will be bootstrapping the 4pt of $SU(N_f)$ adjoint boson bilinears, so besides the $U(N_c)$ scalar gauge theory we also need to consider other theories that contain such operator:

1. Tri-critical QED: It corresponds to the UV fixed point of the scalar QED. It can also be described by Eq. (6), but different from the scalar QED, hitting the tri-critical QED fixed point requires the fine tuning of two singlet operators, i.e., $|\phi|^2$ and $(|\phi|^2)^2$. The relation between the tri-critical QED and scalar QED is similar to the relation between the Gaussian and WF CFT.

2. $SU(N_c)$ QCD: $N_f$ flavor of critcal bosons coupled to a $SU(N_c > 1)$ gauge field.

3. $O(2N_f)^*$:[7] This theory is nothing but replacing the $U(1)$ gauge field of scalar QED in Eq. (6) with a discrete gauge field (e.g. say $Z_N$). It is almost identical to the $O(2N_f)$ WF except only gauge invariant operators are physically allowed in $O(2N_f)^*$. Equivalently, one can also consider branching $O(2N_f)$ into $SU(N_f) \times U(1)$, and only consider the $U(1)$ neutral sector. In this branching, the $O(2N_f)$ symemtric rank-2 traceless tensor becomes the $SU(N_f)$ adjoint.

4. Chern-Simons (CS) gauge theories: In $3d$ one can add a quantized CS term to the $U(1)$ gauge field at any integer level $N$, $N/4\pi\epsilon_{\mu\nu\rho}a_\mu\partial_\nu a_\rho$, leading to a family of parity breaking CFTs [63]. Similarly, one can also consider QCD theories with finite CS terms.

5. Generalized free field (GFF) theory: it is worth noting that there could be different GFFs. One type of GFF (dubbed GFF-A) is made of the $SU(N_f)$ fundamental $\phi_i$, meaning that the $SU(N_f)$ adjoint is constructed by $\phi^i\phi_j$. The other GFF (dubbed GFF-B) is directly made of $SU(N_f)$ adjoint $A^i_j$. One difference between these two GFFs is, the OPE $(\phi^i\phi_j) \times (\phi^k\phi_l)$ in GFF-A contains $\phi^i\phi_j$, while $A^i_j \times A^k_l$ in GFF-B does not contain $A^i_j$.

The last four theories do not have the identical symmetry as the scalar QED, but bootstrapping $SU(N_f)$ adjoint will not be able to tell the difference.[8]

   The decoupling operator we identified in Sec. 2.2 can be used to exclude $SU(N_c > 1)$ gauge theories and GFF-B, while for other theories we need to rely on EOMs. Some consequences of EOMs have already been discussed [36, 37] and been used in the bootstrap analysis [16, 64]. Here we push the idea further, in specific we will discuss 1) the consequence of the EOM of gauge field; 2) high level spectrum due to the EOM. These results will help us to distinguish the scalar QED from its other cousins, particularly the tri-critical QED, $O(2N_f)^*$, and GFF-A.

---

[6]An independent work [61] has a similar analysis for $SO(2)$ gauge theory in the context of $SO(N)$ invariant CFTs.

[7]We adopt the terminology in condensed matter literatures [62].

[8]One can also consider more complicated gauge theories, e.g., critical bosons coupled to a product gauge field $G_1(N_c^1) \times G_2(N_c^2)\cdots$, with $G_i$ to be Lie groups such as $U$, $SU$. The decoupling operator can be used to exclude theories that contain non-Abelian subgroups (gauge group), i.e. $\exists N_c^i > 1$. So the remaining theory one needs to consider has a gauge field $U(1)^m$, which happens to be equivalent to the scalar QED.

Table 1: List of low lying (parity even) primary operators of various theories in $3d$ and the large $N_f$ limit. For notational brevity we omit terms like $-1/N_f \, \delta^i_j |\phi|^2$ for the operators in the adjoint representation. In the table we have $O_1 = (\bar{\phi}^i D_\mu \phi_j)|\phi|^2$, $O_2 = (\bar{\phi}^i \phi_j)\partial_\mu |\phi|^2$, $O_3 = (\bar{\phi}^i \phi_j)(\bar{\phi}^k D_\mu \phi_k)$, and we skip the concrete forms of operators in the last row as it is not illuminating to write them down explicitly.

| | Level | GFF-A | Scalar QED | tri-critical QED | $O(2N_f)^*$ |
|---|---|---|---|---|---|
| Singlet $l=0$ | $\Delta = 1 + O(\frac{1}{N_f})$ | $|\phi|^2$ | None | $|\phi|^2$ | None |
| | $\Delta = 2 + O(\frac{1}{N_f})$ | $(|\phi|^2)^2$ | $\sigma$ | $(|\phi|^2)^2$ | $\sigma$ |
| Adjoint $l=0$ | $\Delta = 1 + O(\frac{1}{N_f})$ | $\bar{\phi}^i \phi_j$ | $\bar{\phi}^i \phi_j$ | $\bar{\phi}^i \phi_j$ | $\bar{\phi}^i \phi_j$ |
| | $\Delta = 2 + O(\frac{1}{N_f})$ | $\bar{\phi}^i \phi_j |\phi|^2$ | None | $\bar{\phi}^i \phi_j |\phi|^2$ | None |
| | $\Delta = 3 + O(\frac{1}{N_f})$ | $\bar{\phi}^i \phi_j(|\phi|^2)^2, \ \bar{\phi}^i \Box \phi_j$ | $\bar{\phi}^i \phi_j \sigma$ | $\bar{\phi}^i \phi_j(|\phi|^2)^2$ | $\bar{\phi}^i \phi_j \sigma$ |
| Singlet $l=1$ | $\Delta = 2$ | $\bar{\phi}^i \overleftrightarrow{D}_\mu \phi_i$ | None | None | $\bar{\phi}^i \overleftrightarrow{D}_\mu \phi_i$ |
| Adjoint $l=1$ | $\Delta = 2$ | $\bar{\phi}^i \overleftrightarrow{D}_\mu \phi_j$ | $\bar{\phi}^i \overleftrightarrow{D}_\mu \phi_j$ | $\bar{\phi}^i \overleftrightarrow{D}_\mu \phi_j$ | $\bar{\phi}^i \overleftrightarrow{D}_\mu \phi_j$ |
| | $\Delta = 3 + O(\frac{1}{N_f})$ | $O_1, O_2, O_3$ | None | $O_1, O_2$ | $O_3$ |
| | $\Delta = 4 + O(\frac{1}{N_f})$ | $\cdots$ | $\cdots$ | $\cdots$ | $\cdots$ |

One can easily analyze the consequence of EOM on the operator spectrum in the perturbative regime, including the large $N_f$ limit, $2 + \epsilon$ limit and $4 - \epsilon$ limit. Here we will consider the large $N_f$ limit. It is known that in the large $N_f$ limit, the Lagrangian of the theory can be written as [44],

$$\mathcal{L} = \sum_{i=1}^{N_f} |(\partial_\mu - iA_\mu)\phi_i|^2 + \sigma |\phi|^2. \tag{6}$$

Here $\sigma$ is a Hubbard-Stratonovich auxiliary field, and the terms $\sigma^2$ and $F^2_{\mu\nu}$ are dropped as they are irrelevant. They are three EOMs (of $\phi$, $\sigma$ and $A_\mu$ respectively),

$$D_\mu D_\mu \phi_i = \sigma \phi_i, \tag{7}$$
$$\bar{\phi}^i \phi_i = |\phi|^2 = 0, \tag{8}$$
$$\bar{\phi}^i \overleftrightarrow{D}_\nu \phi_i = 0. \tag{9}$$

The first two are similar to the EOMs of the WF CFTs, with the difference that the conventional derivative $\partial_\mu$ is replaced by the covariant derivative $D_\mu = \partial_\mu - iA_\mu$. For the brevity of notation we will also write $D_\mu D_\mu = \Box$. The last one is unique for gauge theories.[9]

All the operators of the scalar QED can be built using $\phi_i$, $\sigma$ and $A_\mu$. Except for monopole operators in 3d, all other operators' scaling dimensions are simply the summation of its constituents' scaling dimensions $(\Delta_{\phi_i}, \Delta_\sigma, \Delta_{A_\mu}) = (d/2 - 1, 2, 1)$, up to $1/N_f$ corrections. It is important to note that any operators proportional to the EOM (e.g. $\bar{\phi}^i \phi_j |\phi|^2$ and $(\bar{\phi}^k \overleftrightarrow{D}_\nu \phi_k)\bar{\phi}^i \phi_j$) shall be removed from the operator spectrum.[10] This would then distinguish the scalar QED from its cousins. Table 1 listed the low lying (parity even[11]) primary operators of the scalar

---

[9]Here $\bar{\phi}^i \overleftrightarrow{D}_\nu \phi_i$ stands for $\bar{\phi}^i[(\partial_\nu - iA_\nu)\phi_i] - [(\partial_\nu + iA_\nu)\bar{\phi}_i]\phi_i$.

[10]This was known in the context of large-$N_f$ WF CFTs, and was also discussed in the large-$N_f$ QED theory [65].

[11]For a parity preserving theory (e.g. scalar QED), only parity even operators can appear in the OPE of two parity even scalars (e.g. $SU(N_f)$ adjoint boson bilinear operators).

QED and its cousins, including the GFF-A, tric-critical QED, $O(2N_f)^*$ in the large-$N_f$ limit in 3d. Comparing the scalar QED with its cousins, one can find that the former has several operators missing in specific channels. These missing operators are the consequences of EOMs. For example, in the channel of $SU(N_f)$ singlet $l = 0$, there is no operator in the scalar QED at the level $\Delta = 1 + O(1/N_f)$, as the operator $\bar{\phi}^i \phi_i$ should be deleted due to the EOM of $\sigma$, $\bar{\phi}^i \phi_i = 0$. Similarly, from the EOM of $A_\mu$, we know that in the scalar QED any operator proportional to the gauge current $J^g_\mu = \bar{\phi}^i \overleftrightarrow{D}_\mu \phi_i$ should be absent. This would then distinguish the scalar QED from the $O(2N_f)^*$. Let us also comment on Chern-Simons theories. The operator spectrum of Chern-Simons theories is similar to the scalar QED, however it does not have parity symmetry. This could be used to distinguish the scalar QED from Chern-Simons theories as we will elaborate later.

It is worth emphasizing that even though we analyze EOMs caused missing operators in the perturbative regime, these results are qualitatively correct in the non-pertrubtative regime. Therefore, it is not only necessary but also safe to input these information in a bootstrap study.

## 4 Numerical results

In this section we will switch gear to numerical results. We will study the scalar QED in $2 < d \leq 4$ dimensions and will start with the single correlator of $SU(N_f)$ adjoint operators, $a = \bar{\phi}^i \phi_j - \delta^i_j / N_f |\phi|^2$. The OPE $a \times a$ is,

$$a \times a = S^+ + Adj^+ + A\bar{A}^+ + S\bar{S}^+ + Adj^- + S\bar{A}^- + A\bar{S}^-. \tag{10}$$

Here $S$ and $Adj$ refer $SU(N_f)$ singlet and adjoint. $A\bar{A}$, $S\bar{S}$, $S\bar{A}$, and $A\bar{S}$ are rank-4 tensors with two upper and two lower indices. The naming convention of these representations is rather simple, for example $A\bar{A}$ means that both the upper and lower indices are anti-symmetric, while $S\bar{A}$ means that the lower indices are symmetric and the upper indices are antisymmetric. The upper script $\pm$ means the intermediate channel has even or odd spins. For the bootstrap equations, one can check Ref. [26].

We will denote the low lying scalar operators in the singlet channel as $s, s', \cdots$; scalar operators in the adjoint channel as $a, a' \cdots$; $l = 1$ operators in the adjoint channel as $J_\mu, J'_\mu, \cdots$. Besides the single correlator of $a$, we will also present some results of mix correlators of $a$ and $s$. We note that $a$ appears in the OPE of $a \times a$, so we impose this condition in all the numerics, for example we require that all the scalars in the adjoint channel should be no smaller than $\Delta_a$. Physically this gap condition does not introduce any assumption to the CFT spectrum, but it does modify the numerical bounds significantly. Most results are calculated with $\Lambda = 27$ (the number of derivatives included in the numerics) unless stated otherwise.

Before going to details, we will summarize some known results about the low lying spectrum of the scalar QED. In 3d, the large-$N_f$ calculation [44, 66] gives

$$\Delta_a = 1 - \frac{48}{3\pi^2 N_f} + O(1/N_f^2), \tag{11}$$

$$\Delta_s = 2 - \frac{144}{3\pi^2 N_f} + O(1/N_f^2). \tag{12}$$

From $2 + \epsilon$ expansion [38–40], one has

$$\Delta_a = \epsilon - \frac{2}{N_f} \epsilon^2 + O(\epsilon^3), \tag{13}$$

$$\Delta_s = 2 - \frac{2}{N_f} \epsilon^2 + O(\epsilon^3). \tag{14}$$

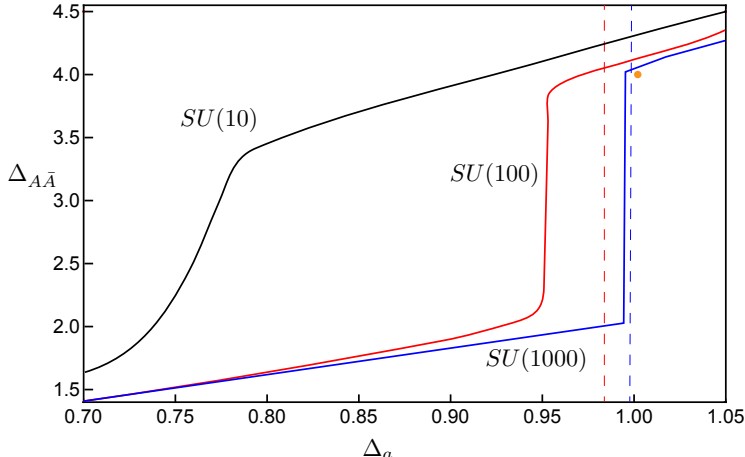

Figure 1: The numerical bounds of the lowest operator in the $A\bar{A}$ channel for $SU(10)$, $SU(100)$, and $SU(1000)$ CFTs in $d = 3$. The dashed line corresponds to the large$-N_f$ results of $\Delta_a$ for the $N_f = 100$ and $N_f = 1000$ scalar QED. The orange circle corresponds to $(\Delta_a, \Delta_{A\bar{A}}) = (d - 2, 2(d - 2) + 2)$.

It is also worth noting the tri-critical QED in $3d$ has [44],

$$\Delta_a = 1 - \frac{64}{3\pi^2 N_f} + O(1/N_f^2), \tag{15}$$

$$\Delta_s = 1 + \frac{128}{3\pi^2 N_f} + O(1/N_f^2). \tag{16}$$

Other results of spectrum will be discussed below when needed.

## 4.1 Kinks of the $A\bar{A}$ bound

As we discussed in Sec. 2.2, the lowest operator in the $A\bar{A}$ channel of non-Abelian gauge theories becomes decoupled in the abelian gauge theories (e.g. scalar QED, tri-critical QED, $O(2N_f^*)$), so it is natural to bound $A\bar{A}$ channel gap $\Delta_{A\bar{A}}$ to see if this operator decoupling can be detected. Concretely, for abelian gauge theories (and GFF-A) we have

$$\Delta_{A\bar{A}} = 2(d - 2) + 2 + O(1/N_f), \tag{17}$$

while for non-Abelian gauge theories (and GFF-B) we have

$$\Delta_{A\bar{A}} = 2(d - 2) + O(1/N_f). \tag{18}$$

Fig. 1 shows the numerical bounds of $\Delta_{A\bar{A}}$ in $3d$. The numerical bounds show clear kinks for different $N_f$'s, and the kink evolves into a vertical jump from $(\Delta_a, \Delta_{A\bar{A}}) = (1, 2)$ to $(\Delta_a, \Delta_{A\bar{A}}) = (1, 4)$ as $N_f \to \infty$. The appearance of the $A\bar{A}$ kink can be ascribed to the decoupling operator theorem of Abelian gauge theories we discussed above. In particular, in the large-$N_f$ limit the Abelian gauge theories are living in the space after the jump, while the non-Abelian gauge theories may live in the space before the jump.

We note that this family of kinks is very similar to the non-WF kinks of $O(N)$ theories [33]. In particular, in 2d the $O(4)$ non-WF kink exactly corresponds to the $SU(2)_1$ WZW CFT. Given that the WZW CFTs' null operators can be viewed as gauge theories' decoupling operators, it is very tempting to conjecture that the $A\bar{A}$ kinks here correspond to the scalar QED. A careful analysis from both the numerical and theoretical perspective suggests that the $A\bar{A}$ kink is

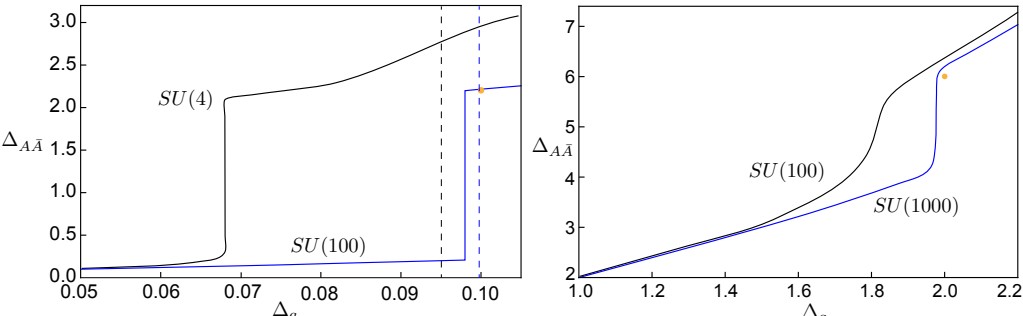

Figure 2: The numerical bounds of the lowest operator in the $A\bar{A}$ channel for $SU(N_f)$ CFTs in $d = 2.1$ (left) and $d = 4$ dimensions (right). The dashed line in the left corresponds to the $2 + \epsilon$ results of $\Delta_a$ for $N_f = 4$ and $N_f = 100$ scalar QED. The orange circle corresponds to $(\Delta_a, \Delta_{A\bar{A}}) = (d - 2, 2(d - 2) + 2)$.

unfortunately not the scalar QED. Although the $1/N_f$ correction of $\Delta_{A\bar{A}}$ is unknown, we can compare $\Delta_a$ of the kinks with the large-$N_f$ results Eq. (11). In Fig. 1 we also plot large-$N_f$ $\Delta_a$ of $SU(100)$ and $SU(1000)$ scalar QED, which shows considerably large discrepancies to the kinks. Take a closer look at the data, the $SU(100)$ kink sits around $\Delta_a \approx 0.953$, while the large $N_f$ results gives $\Delta_a \approx 0.984$. The discrepancy between these two numbers is around $3/N_f$. Similarly, this is also the case for $SU(1000)$, which has $\Delta_a \approx 0.995$ and $\Delta_a \approx 0.998$ for the kink and large $N_f$, respectively. This large discrepancy does not seem to be caused by a numerical convergence issue, as the differences of $\Delta_a$ between $\Lambda = 19, 27, 35$ are small. Theoretically, it is indeed easy to convince oneself that the $A\bar{A}$ kink cannot be the scalar QED. That is because the tri-critical QED also has $\Delta_{A\bar{A}} = 2(d - 2) + 2 + O(1/N_f)$, and its $\Delta_a$ (Eq. (15)) is smaller than that of the scalar QED (Eq. (11)). As a side note, in theory the $A\bar{A}$ kink could be the tri-critical QED, but numerically it does not seem be so as their large-$N_f$ $\Delta_a$'s also have more than $2/N_f$ discrepancy from the numerical kink.

We have also studied $A\bar{A}$ bound in other dimensions (see Fig. 2). We find that the $A\bar{A}$ kinks still exist in $2 < d \leq 4$ dimensions, and the kink approaches $(\Delta_a, \Delta_{A\bar{A}}) = (d-2, 2(d-2)+2)$ as $N_f \rightarrow \infty$. It is worth noting that, in 4d for $N_f \neq \infty$ the $A\bar{A}$ kink does not sit at $(\Delta_a, \Delta_{A\bar{A}}) = (2, 6)$. This again suggests this kink should not be identified as the scalar QED or tri-critical QED, as both of them shall flow to the Gaussian fixed point in 4d. In $d = 2.1$ dimensions, the $A\bar{A}$ kinks again have large deviations to the $2 + \epsilon$ results of the scalar QED.

Therefore, the single correlator can capture the essential physics of the $A\bar{A}$ decoupling from non-Abelian gauge theories to Abelian gauge theories. However, the $A\bar{A}$ kink does not correspond to any known CFT. This result inspires us that, instead of bounding $\Delta_{A\bar{A}}$ we can impose a gap in the $A\bar{A}$ channel to exclude all the non-Abelian gauge theories. We will pursue this in the remaining part of this paper.

### 4.2 Scalar QED islands in 3 dimensions

Interestingly, by imposing gaps $\Delta_{A\bar{A}} \geq 3$, $\Delta_{J'_\mu} \geq 3.1$, and $\Delta_{S\bar{S}'} \geq 3.1$ in the operator spectrum, we are able to obtain bootstrap islands of scalar QED in $d = 3$ dimensions by scanning the $\Delta_a$-$\Delta_{S\bar{S}}$ space, as shown in Fig. 3. These three gaps are pretty mild compared to the real gaps of scalar QED, and they have very clear physical meanings: 1) $\Delta_{A\bar{A}} \geq 3$ serves to exclude non-Abelian gauge theories (i.e. QCD); 2) $\Delta_{J'_\mu} \geq 3.1$ serves to exclude $O(2N_f)^*$ and Chern-Simons theories; 3) $\Delta_{S\bar{S}'} \geq 3.1$ serves to exclude tri-critical QED and GFF-A. Table 2 gives a summary of imposed gaps and physical gaps of various theories.

The numerics seems to converge better for large $N_f$. For instance, for $N_f = 1000$ we can

Table 2: The imposed spectrum gaps for the scalar QED island in Fig. 3 and the physical gaps of different theories. Most physical gaps have already been analyzed above in Table 1. For Chern-Simons theories, we have $J'_\mu = a \cdot \varepsilon_{\mu\nu\rho} F_{\nu\rho}$, whose scaling dimension is $\Delta = 3 + O(1/N_f)$. In the scalar QED such operator also exists, but it is a parity odd operator, hence will not appear in the $a \times a$ OPE.

|  | Gap imposed | Scalar QED | Tri-critical QED | GFF-A | $O(2N_f^*)$ | QCD | Chern-Simons |
|---|---|---|---|---|---|---|---|
| $\Delta_{A\bar{A}}$ | 3 | $4 + O(\frac{1}{N_f})$ | $4 + O(\frac{1}{N_f})$ | $4 + O(\frac{1}{N_f})$ | $4 + O(\frac{1}{N_f})$ | $2 + O(\frac{1}{N_f})$ | $4 + O(\frac{1}{N_f})$ |
| $\Delta_{J'_\mu}$ | 3.1 | $4 + O(\frac{1}{N_f})$ | $3 + O(\frac{1}{N_f})$ | $3 + O(\frac{1}{N_f})$ | $3 + O(\frac{1}{N_f})$ | $4 + O(\frac{1}{N_f})$ | $3 + O(\frac{1}{N_f})$ |
| $\Delta_{S\bar{S}'}$ | 3.1 | $4 + O(\frac{1}{N_f})$ | $3 + O(\frac{1}{N_f})$ | $3 + O(\frac{1}{N_f})$ | $4 + O(\frac{1}{N_f})$ | $4 + O(\frac{1}{N_f})$ | $4 + O(\frac{1}{N_f})$ |

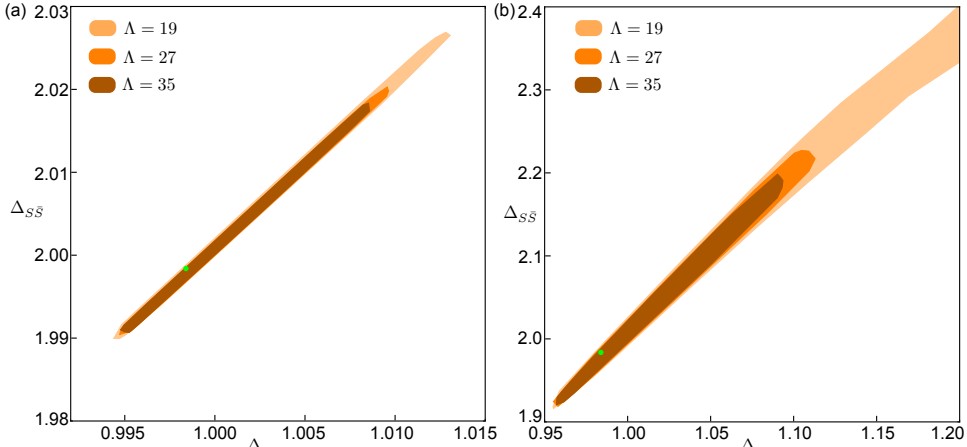

Figure 3: Scalar QED islands in $3d$: (a) $N_f = 1000$, (b) $N_f = 100$. The allowed regions (shaded) are obtained by imposing gaps in operator spectrum summarized in Table 2. The green circles are the large$-N_f$ results of scalar QED, with $\Delta_{S\bar{S}} = 2 - \frac{48}{3\pi^2 N_f}$ [44] and $\Delta_a$ in Eq. (11).

get an island with $\Lambda = 19$, while for $N_f = 100$ $\Lambda = 19$ does not yield an island. Moreover, for $N_f = 50$ we are not able to obtain an island up to $\Lambda = 35$. It could be that for small $N_f$ the scalar QED island still exist for large $\Lambda$, which, however, is beyond our computational power. Up to $\Lambda = 35$, the size of the island is proportional to $1/N_f$. It is unknown if the islands will further shrink as $\Lambda$ increases. It is worth remarking that in $O(N)$ WF bootstrap, the island of $O(3)$ WF from two operators mix also shrinks rather slowly with $\Lambda$ (in a similar rate as Fig. 3) [16], but three operators mix could drastically shrink the island [24]. It will be very interesting to mix $a$ with $S\bar{S}$ to see how small the scalar QED island will shrink to, and to see if a scalar QED island will also exist for small $N_f$ using an accessible $\Lambda$.

The appearance of scalar QED islands strongly advocates our proposed recipes for bootstrapping gauge theories. We also remark that, a basic requirement to isolate a CFT of interest into an island is to impose a set of gaps that exclude other crossing symmetric theories. From this perspective, the single correlator bootstrap could also do the job of isloating a CFT as long as it can access a set of such gaps, and it has also been demonstrated for $O(N)$ WF [67]. The mixed correlator bootstrap, on the other hand, is certainly more powerful than the single correlator bootstrap for several reasons. Firstly, the mixed correlator can access new decoupling/missing operators that are absent in the single correlator bootstrap. For instance, the mixed correlator of $O(N)$ vector and singlet can detect the missing operator in the $O(N)$ vector channel, i.e., $\phi|\phi|^2$ [16]. Secondly, the mixed correlator has stronger constraining power and better numerical convergence.

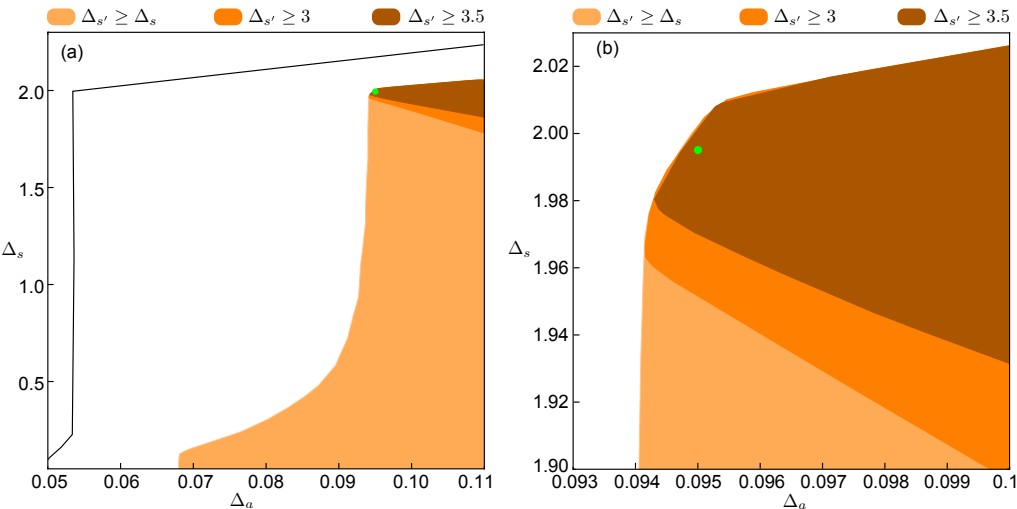

Figure 4: The numerical bounds of $SU(4)$ singlet $\Delta_s$ of $SU(4)$ CFTs in $d = 2.1$ dimension. The allowed regions (shaded) are obtained with the gaps $\Delta_{A\bar{A}} \geq 2(d-2)+1$ and $\Delta_{s'}$ itemized in the figure. The green circle corresponds to scaling dimensions of the scalar QED from epsilon expansion $(\Delta_a, \Delta_s) \approx (0.095, 1.995)$. The right panel is the zoomed in plot of the left panel. In the left panel, we also plot the numerical bounds (i.e. black curve) that does not have any gap imposed.

## 4.3 Scalar QED kinks and islands in $2 + \epsilon$ dimensions

The above discussed scalar QED islands in the $\Delta_a$-$\Delta_{S\bar{S}}$ space also exist in $2 + \epsilon$ dimensions. Moreover, in $2 + \epsilon$ limit the numerical convergence becomes much faster, and we are able to obtain bootstrap island for small $N_f$ (e.g. $N_f = 4$) with a small $\Lambda$. We will not repeat such discussions here. It turns out illuminating to study $\Delta_s$ bound in $2 + \epsilon$ dimensions, as will be detailed in this section.

We first add a mild gap $\Delta_{A\bar{A}} \geq 2(d-2)+1$ in the $A\bar{A}$ channel that excludes all the non-Abelian gauge theories (as well as the GFF-B). Having excluded all the non-Abelian gauge theories, the remaining cousins of the scalar QED that are consistent with the crossing symmetry are the tri-critical QED, GFF-A and $O(2N_f)^*$. As we discussed in Sec. 3 the difference between the scalar QED and tri-critical QED/GFF is that, the former contains $\phi^4$ interactions, while the latter does not. This difference is similar to the difference between the WF CFT and GFF/Gaussian. For the $O(N)$ WF CFT, it is well known that one can detect it as a kink that is above the GFF by bounding the $O(N)$ singlet [16]. So one may expect that the scalar QED would appear as a kink if one bounds the $SU(N_f)$ singlet $\Delta_s$.

Fig. 4 shows the numerical bounds of $\Delta_s$ of $N_f = 4$,[12] which has a kink that is close to the $2 + \epsilon$ expansion results of scalar QED.[13] We further impose a gap in the second low lying singlet $\Delta_{s'} \geq 3, 3.5$ and scan the feasible region of $\Delta_s$. The $\Delta_{s'}$ gap carves out a large region, leaving a sharp tip where the scalar QED sits in. This phenomenon is similar to that of Ising CFT, for which imposing further constraints will carve the feasible region into a small island [14]. Below we will show that the feasible region of scalar QED also shrinks to an island in the $\Delta_a$-$\Delta_s$ space with proper conditions imposed.

It is good to pause here to elaborate a bit more on the philosophy of imposing gap conditions in bootstrap calculations. As we have explained, in many cases, in particular for gauge theories, it is necessary to impose gaps in order to exclude other theories that are also con-

---

[12]The results of different $N_f$'s are rather similar, so we just choose $N_f = 4$ as a representative one.

[13]The discrepancy is of order $O(\epsilon^3)$ and $O(\epsilon^2)$ for $\Delta_a$ and $\Delta_s$, respectively.



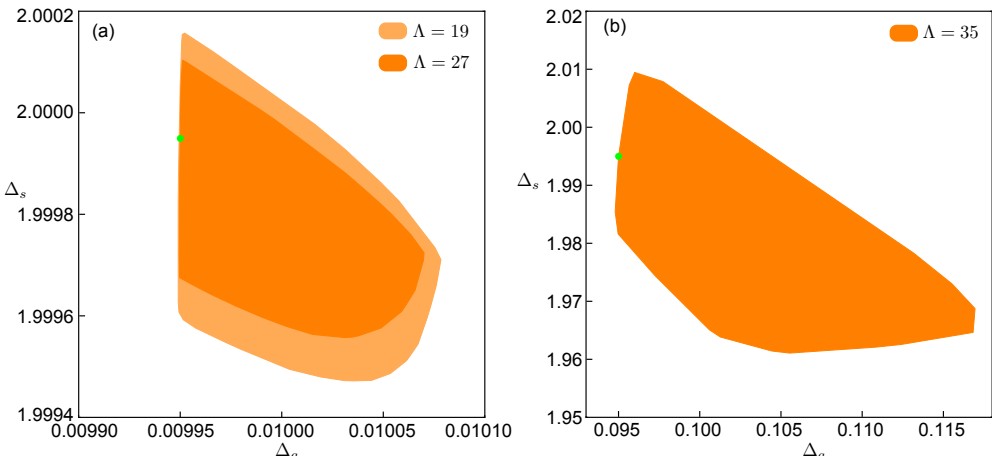

Figure 5: The islands of the scalar QED with $N_f = 4$. The green circles mark $2 + \epsilon$ results of the scalar QED. (a) $d = 2.01$ dimensions: the feasible regions are obtained from the single correlator. (b) $d = 2.1$ dimensions: The feasible region is obtained from the $a, s$ mixed correlator.

sistent with crossing equations. On the other hand, in bootstrap calculations it is common that imposing gaps will carve out feasible regions, possibly leaving a kink on the numerical bounds. Sometimes, the kink is floating, namely it is moving as the gap changes (see appendix for more details). Such floating kink does not unambiguously correspond to an isolated CFT. On the practical side, it is hard to extract useful information about the physical theory from a floating kink unless one already has the knowledge of precise values of the gaps. In contrast, the kink in Fig. 4 is stable, namely it does not move as long as the gap ($\Delta_{A\bar{A}}$) is in a finite window. We have explicitly checked that the kink and numerical bounds are almost identical for different values of gap, i.e. $\Delta_{A\bar{A}} \geq 2(d-2) + 1$ and $\Delta_{A\bar{A}} \geq 2(d-2) + 1.5$. On the other hand, if one removes the $\Delta_{A\bar{A}}$ gap, $\Delta_s$ bound gets modified significantly (the black curve in Fig. 4(a)): The scalar QED kink disappears, but there is one kink close to the unitary bound (of $\Delta_a$) which is likely to be a WF type theory. These results justify our decoupling operator based recipes for bootstrapping gauge theories, in specific the $\Delta_{A\bar{A}}$ gap is serving to exclude all the non-Abelian gauge theories.

We also remark that there is a vertical kink on the leftmost feasible region. It corresponds to the $\Delta_{A\bar{A}}$ jump shown in Fig. 1-2. It is noticeable that $\Delta_s$ is pretty small in this region, supporting again that the $\Delta_{A\bar{A}}$ kink (jump) cannot be the scalar QED. It will be interesting to know if the tri-critical QED lives in any special region (e.g. the leftmost kink) of the numerical bounds.

To get an island of the scalar QED, we need to find conditions to exclude all its cousins. Similar to Table 2, we impose the following mild gaps in the operator spectrum,

$$\Delta_{A\bar{A}} \geq 2d - 3, \Delta_{s'} \geq 3, \Delta_{J'_\mu} \geq 2d - 2.8, \Delta_{S\bar{S}} \geq \Delta_a, \tag{19}$$

and we successfully isolate the scalar QED into a small island (in the $\Delta_a - \Delta_s$ space) with the single correlator in $d = 2.01$ dimensions, as shown in Fig. 5(a). The first three gaps have very clear physical meanings, they serve to exclude non-Abelian gauge theories, tri-critical QED/GFF, and $O(2N_f^*)$. The last gap $\Delta_{S\bar{S}}$ is rather mysterious, we do not have a clear idea what theory does it exclude. Removing any of these four gaps, the scalar QED will not be isolated to an island any more. Somewhat surprisingly, by increasing the dimensions slightly, say $d = 2.1$, the single correlator can not isolate an island any more. The mixed correlator can

still yield an island with a high $\Lambda = 35$ [14] and more aggressive (but still physical) gap conditions (Fig. 5(b)), i.e., $\Delta_{A\bar{A}} \geq 2(d-2)+1$, $\Delta_{s'} \geq 3.5$, $\Delta_{J'_\mu} \geq d + 0.5$, $\Delta_{S\bar{S}} \geq \Delta_a$, $\Delta_{a'} \geq \Delta_a + 1.5$.

The appearance of scalar QED kinks and islands in $d = 2 + \epsilon$ dimension again advocate our proposed recipes for bootstrapping critical gauge theories. These nice results, however, do not sustain to $d = 3$ dimensions. More detailed numerical observations and discussions can be found in Appendix B.

## 5 Conclusion and outlook

We have introduced the notion of decoupling operators of critical gauge theories in dimensions $d > 2$. The decoupling operator is the higher dimensional reminiscent of null operators of 2d WZW CFTs, and it can efficiently detect the rank of the gauge group. Based on the information of decoupling operators, one can then impose gap conditions in bootstrap calculations to isolate gauge theories of interest from other theories. As an illustrative example, we study a prototypical critical gauge theory, i.e., the scalar QED. We firstly identified the concrete decoupling operators of the scalar QED, and then showed how to use them in a bootstrap study.

In both the 3d large-$N_f$ limit and the $d = 2 + \epsilon$ limit, we have successfully obtained kinks as well as islands of the scalar QED, by imposing mild gap conditions inspired by the physics of decoupling operators and EOMs. We shall remark that, even though these two limits can be accessed using perturbative expansions, our bootstrap calculations do not rely on any of these perturbative results. The gap conditions we imposed are very mild that are likely to hold for any $N_f$ in 3d. The success of bootstrap calculations, however, does not sustain to the most interesting case, i.e., small $N_f$ in 3d. The failure for small $N_f$ in 3d might be due to the poor numerical convergence. It is possible that the mixed correlator bootstrap between $a$ and $S\bar{S}$ will improve the numerical convergence significantly and solve the long-standing problem regarding the properties of small $N_f$ scalar QED in 3d. We will leave this for the future study.

One interesting question is what does the $A\bar{A}$ kink in Fig. 1 and Fig. 2 correspond to? This family of kinks shares a lot of similarities as the vertical jump in the bound of rank-2 symmetric tracless tensor of the $O(N)$ theories (this kink was dubbed non-WF kink) [33]. Also a similar kink was recently observed in bootstrapping $O(N)$ rank-2 symmetric traceless tensor [61]. We believe these kinks may have similar physical mechanisms. They could either be unknown CFTs or artifacts of numerical bootstrap. Even if they are numerical artifacts, the crossing symmetric solution at the kink may have certain relations to gauge theories, given that they are close to gauge theories in the parameter space. Understanding them may help to eventually solve the gauge theories in 3d.

We have showed how to use the decoupling operator in the $A\bar{A}$ channel to bootstrap $U(1)$ gauge theories. In a similar fashion, one can bootstrap a non-Abelian gauge theory with a specific gauge group $U(N_c = m)$ by using the decoupling operators in the antisymmetric representations $A^{[f_1,\cdots,f_{n+1}]}_{[f_{n+2},\cdots,f_{2n+2}]}$ of $SU(N_f)$ with $n \leq m$. For example, in the channel $A^{[f_1,\cdots,f_{m+1}]}_{[f_{m+2},\cdots,f_{2m+2}]}$ the lowest operator of different gauge theories will have distinct scaling dimensions: 1) the $U(N_c > m)$ gauge theories have $\Delta = (m+1)(d-2) + O(1/N_f)$; 2) the $U(N_c = m)$ gauge theories have $\Delta = (m+1)(d-2) + 2 + O(1/N_f)$; 3) the $U(N_c < m)$ gauge theories have $\Delta \geq (m+1)(d-2) + 4 + O(1/N_f)$. We also remark that as a concrete example we analyzed decoupling operators of theories with a $U(N_c)$ gauge field coupled to bosons. It is straightforward to generalize to other gauge groups (e.g. $SU(N_c), SO(N_c), USp(2N_c)$) as well as fermions coupled to gauge fields. It will be interesting to try our decoupling operator based recipes to tackle other gauge theories. In particular, exciting progress might be made by using advanced

---

[14] $\Lambda = 27$ does not produce an island.

bootstrap techniques such as mixing spinning operators [68].

On the phenomenological level the decoupling operators of gauge theories share several similarities with the null operators of 2d WZW CFTs. As detailed in Sec. 2.1 the null operators of $SU(N)_k$ WZW CFTs can be even considered as decoupling operators of 2d $U(k)$ gauge theories. In the context of 2d CFTs the null operator has important applications, e.g. they can act as differential operators that annihilate correlation functions. It is an open question whether a similar application also exists for the decoupling operators of gauge theories in a general dimension. The progress might be made by looking for an exact interpolation between gauge theories with different gauge groups, which is similar to the interpolation between WZW CFTs with different WZW levels.

# Acknowledgements

YCH would like to thank Chong Wang and Liujun Zou for the stimulating discussions and collaborations on 3d WZW models, and Zheng Zhou for the discussions on the large$-N_f$ equation of motion, which benefit current work. We thank Slava Rychkov for his critical reading of our manuscript and for his various suggestions. Research at Perimeter Institute is supported in part by the Government of Canada through the Department of Innovation, Science and Industry Canada and by the Province of Ontario through the Ministry of Colleges and Universities. This project has received funding from the European Research Council (ERC) under the European Union's Horizon 2020 research and innovation programme (grant agreement no. 758903). The work of J.R. is supported by the DFG through the Emmy Noether research group The Conformal Bootstrap Program project number 400570283. The numerics is solved using SDPB program [69] and simpleboot (https://gitlab.com/bootstrapcollaboration/simpleboot). The computations in this paper were run on the Symmetry cluster of Perimeter institute, and on the EPFL SCITAS cluster funded by the Swiss National Science Foundation under grant no. PP00P2-163670. NS would like to thank his parents for support during the COVID-19 pandemic. NS would like to thank the hospitality of Institute of Physics Chinese Academy of Sciences and the Center for Advanced Study, Tsinghua University while part of the work was finished.

# A   3d WZW models and gauge theories

In this appendix, we will discuss some examples that show direct connections between WZW CFTs and 3d gauge theories. The physics discussed here is not new, it is the recollection of the results in Ref. [59, 60].

Despite of the pure algebraic definition, 2d WZW CFTs also have a Lagrangian formulation, namely a non-linear sigma model (NL$\sigma$M) on a (Lie) group manifold $G$ ($SU(N)$, $USp(2N)$, etc.) supplemented with a level $k$ WZW term [56],

$$\mathcal{L} = \frac{1}{4a^2} \int d^2x \, \text{Tr}(\partial^\mu g^{-1})(\partial_\mu g) + k \cdot \frac{i}{24\pi} \epsilon_{\mu\nu\rho} \int_B d^3x \text{Tr}((\hat{g}^{-1}\partial^\mu \hat{g})(\hat{g}^{-1}\partial^\nu \hat{g})(\hat{g}^{-1}\partial^\rho \hat{g})). \quad (20)$$

$g$ is a matrix field valued in a unitary presentation of the Lie group. The first term is the ordinary kinetic term of NL$\sigma$M, the second term is the WZW term defined in the 3-dimensional extended space. $k$ is quantized and corresponds to the homotopy class $\pi_3(G) = \mathbf{Z}$. One shall also have $\pi_2(G) = \mathbf{0}$ in order for the WZW term to be well defined. The Lagrangian has a conformal fixed point (i.e. WZW CFT) at a finite coupling strength.

It is straightforward to generalize the WZW Lagrangian to a higher dimension. In 3d a non-trivial WZW term requires the target space $G$ to satisfy $\pi_4(G) = \mathbf{Z}$ and $\pi_3(G) = \mathbf{0}$. There are several target spaces, including Grassmannian and Stiefel manifold (e.g. $SO(N)/SO(4)$), satisfying this requirement. One important difference in $3d$ is that the NL$\sigma$M is non-renormalizable, making it hard to analyze.[15] Nevertheless, it was argued that [59],[16] there are three fixed points as the coupling strength $a^2$ increases from 0:

1. An attractive fixed point of spontaneous symmetry breaking (SSB) phase at $a^2 = 0$. The ground state manifold is the target space of NL$\sigma$M.

2. A repulsive fixed point of order-disorder phase transitions.

3. An attractive conformal fixed point preserving all the symmetries.

The last attractive conformal fixed point is the $3d$ version of the 2d WZW CFT, while the first two fixed points merge into the Gaussian fixed point in 2d.

Ref. [59] studied such $3d$ WZW models on the Stiefel manifold, here we discuss a simpler situation–the 3d Grassmannian $U(2N)/(U(N) \times U(N))$ WZW models [60]. In particular, we will argue that the Grassmannian WZW models have simple UV completions, i.e., Dirac fermions coupled to a gauge field.

The UV completion of the 3d leve-$k$ $U(2N)/(U(N) \times U(N))$ WZW model is the QCD$_3$-Gross-Neveu model,

$$\mathcal{L} = \sum_{i=1}^{2N} \bar{\psi}^i \gamma_\mu (\partial_\mu - i\alpha_\mu)\psi_i + \lambda \phi_j^i \left( \bar{\psi}_j \psi_i - \frac{1}{2N} \delta_i^j \bar{\psi}\psi \right)$$
$$+ \mathrm{Tr}((\partial\phi)^2) + m\mathrm{Tr}(\phi^2) + u_1 \mathrm{Tr}(\phi^4) + u_2 (\mathrm{Tr}(\phi^2))^2 .$$

Here $\alpha_\mu$ is a $SU(k)$ gauge field, $\psi_i$ Dirac fermions are in the $SU(k)$ fundamental presentation. $\phi_j^i$ is a bosonic field in the $SU(2N)$ adjoint representation, and it is coupled to the adjoint mass term of the Dirac fermions.

The QCD$_3$-Gross-Neveu model model has three fixed points, corresponding to a SSB phase with ground state manifold $U(2N)/(U(N) \times U(N))$, QCD$_3$-Gross-Neveu CFT, and QCD$_3$ CFT. The QCD$_3$-Gross-Neveu CFT fixed point is unstable, and will flow to either the SSB or the QCD$_3$ CFT depending on the sign of $m\mathrm{Tr}(\phi^2)$. This phase diagram coincides with that of $U(2N)/(U(N) \times U(N))$ WZW models. In the SSB phase of the QCD$_3$-Gross-Neveu model, one can define a NL$\sigma$M model on the target space $U(2N)/(U(N) \times U(N))$. In the SSB phase, the Dirac fermions are gapped, integrating out of them will generate a level-$k$ WZW term [70]. The level $k$ (instead of 1) comes from the color multiplicity of Dirac fermions due to the $SU(k)$ gauge field. Therefore, we have proved that the SSB fixed point of the QCD$_3$-Gross-Neveu model and the level-$k$ $U(2N)/(U(N) \times U(N))$ WZW are dual to each other.

Given that phase diagrams of two models match and the SSB phase of two models are dual, it is natural to conjecture that the QCD$_3$-Gross-Neveu model is the UV completion of 3d WZW model on $U(2N)/(U(N) \times U(N))$ manifold. In particular,

- *The IR conformal fixed point of the 3d level$-k$ $U(2N)/(U(N) \times U(N))$ WZW model is dual to the QCD$_3$ CFTs with $N_f = 2N$ Dirac fermions coupled to a $SU(k)$ gauge field.*

---

[15]A theory being non-renormalizable does not necessarily mean it is non-sensible. For the context of NL$\sigma$M, we know that it can describe the WF CFTs although it is non-renormalizable in $d > 2$ dimensions.

[16]Ref. [59] studied Stiefel manifold, but it should be readily generalized to other manifold.

There is an interesting sanity check for this duality. The Grassmannian $U(2N)/(U(N) \times U(N))$ has a nontrivial $\pi_2 = \mathbf{Z}$ leading to Skyrmion operators. The Skyrmion is either a boson or fermion depending on the evenness and oddness of $k$ [60]. The Skyrmion can be identified as the baryon operator of the $SU(k)$ gauge theory, whose statistics also depends on $k$.

Similarly, one can derive,

- *The IR conformal fixed point of the 3d level$-k$ $SO(2N)/(SO(N) \times SO(N))$ WZW model is dual to the QCD$_3$ CFTs with $N_f = 2N$ Dirac fermions coupled to a $SO(k)$ gauge field.*

- *The IR conformal fixed point of the 3d level$-k$ $USp(4N)/(USp(2N) \times USp(2N))$ WZW model is dual to the QCD$_3$ CFTs with $N_f = 2N$ Dirac fermions coupled to a $USp(2k)$ gauge field.*

# B   More numerical data

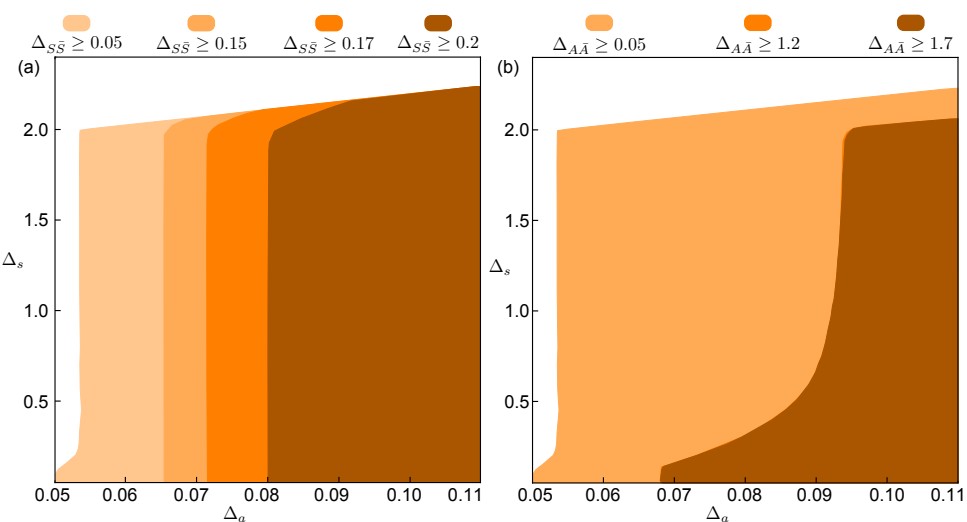

Figure 6: Floating kinks versus stable kinks in $d = 2.1$ dimensions with a global symmetry $SU(4)$. (a) Example of floating kinks. (b) Example of stable kinks. The feasible regions calculated with the gap conditions $\Delta_{A\bar{A}} \geq 1.2$ and $\Delta_{A\bar{A}} \geq 1.7$ are almost identical to each other.

In this appendix we will provide more detailed numerical data, and most of the data will focus on $2 + \epsilon$ dimensions.

Firstly, let us briefly comment on floating kinks and stable kinks. As we have explained in the main text, the floating kink means the kink is moving as the imposed gap changes, while the stable kink means that the kink is not moving as long as the imposed gap lies in a finite window. Fig. 6 shows a concrete comparison between floating kinks and stable kinks. The floating kinks in Fig. 6(a) clearly show dependence on the values of $\Delta_{S\bar{S}}$ gap. In contrast, the stable kinks in Fig. 6(b) show little dependence on the value of the gap.

To have a more intuitive idea about the magic of EOMs, we have investigated how the bound of $\Delta_{J'_\mu}$ evolves with $\Delta_a$. As shown in Fig. 7, the scalar QED sits at a sharp spike, which is well separated from $O(2N_f)^*$. This is the consequence of EOM of gauge field, as discussed in Table 1. The sharp spike also explains why the gap of $\Delta_{J'_\mu}$ helps to isolate the scalar QED into an island. Another noteworthy observation is that convergence quickly becomes difficult as the dimension $d$ increases slightly. In $d = 2.01$ dimensions (Fig. 7(a)) $\Delta_{J'_\mu}$ has a sharper

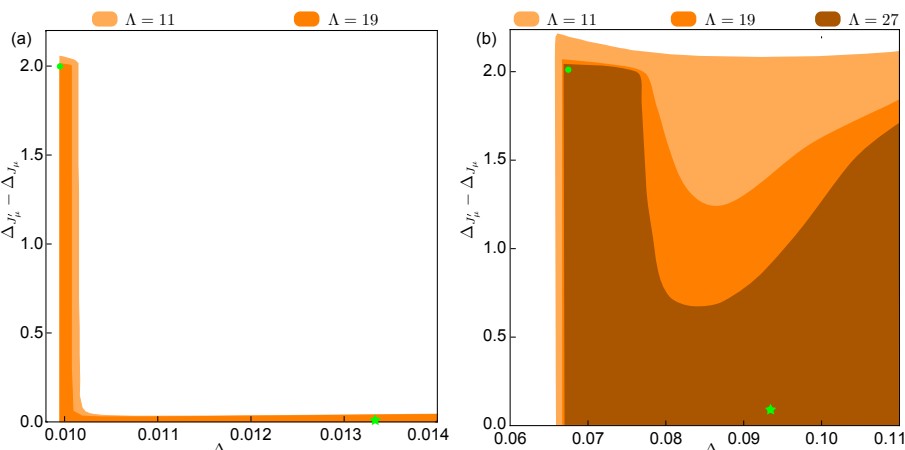

Figure 7: The numerical bounds of $\Delta_{J'_\mu}$ of $SU(4)$ CFTs with gap $\Delta_{A\bar{A}} \geq 2(d-2)+1$, $\Delta_s \geq 1$, $\Delta_{S\bar{S}} \geq \Delta_a$ in (a) $d = 2.01$ dimensions, (b) $d = 2.07$ dimensions. The shaded regions are allowed regions. The green circles mark the scalar QED and the green stars mark $O(2N_f)^*$, up to $O(\epsilon^3)$ and $O(\epsilon)$ corrections for $\Delta_a$ and $\Delta_{J'_\mu}$, respectively.[17]

spike for a small $\Lambda = 11$, and a larger $\Lambda = 19$ does not improve the bound significantly. In contrast, in $d = 2.07$ dimensions $\Lambda = 11$ does not produce a spike at all, while the spike shows up weakly for $\Lambda = 19$ and becomes sharper for $\Lambda = 27$. Moving to a higher dimension (e.g. $d = 2.1$) the spike does not show up even for $\Lambda = 27$ (the feasible region looks similar to that of $d = 2.07$ with $\Lambda = 11$ in Fig. 7(b)). This also explains why the single correlator does not produce an island in $d = 2.1$ dimensions for $N_f = 4$. We also want to remark that the convergence becomes easier for a larger $N_f$, e.g. $\Delta_{J'_\mu}$ still has a spike in $d = 2.3$ dimensions for $SU(100)$ with $\Lambda = 19$. This also agrees that in $3d$ we are able to obtain islands in the $\Delta_S - \Delta_{S\bar{S}}$ space for large $N_f$ (i.e. Fig. 3).

Finally, let us investigate how the scalar QED kinks evolve as we approach $d = 3$ dimensions. In a given dimension there exists a critical $N_f^*(d)$ below which the scalar QED will lose its conformality. It remains an open question about the precise value of $N_f^*$ in $d = 3$ dimensions. To avoid the unnecessary complexity, we choose a large $N_f = 100$ to monitor how the scalar QED kink evolves as the dimension increases.

Fig. 8 shows the numerical bounds of $\Delta_s$ in $d = 2.1, 2.4, 2.7, 3$ dimensions. In every plot there is a sharp vertical kink on the leftmost side of the feasible region. This kink is the $A\bar{A}$ kink discussed in Sec. 4.1, and does not correspond to the scalar QED. In $d = 2.1$ dimensions (Fig. 8(a)), similar to $N_f = 4$ in Fig. 4 the numerical bound has a sharp kink that is close to the $2 + \epsilon$ result $(\Delta_a, \Delta_s) = (0.0998, 1.9998)$ of the scalar QED. As $d$ increases, the scalar QED kink becomes weak in $d = 2.4$ (Fig. 8(b)), and finally becomes invisible in $d = 2.7$ (Fig. 8(c)) and $d = 3$ dimensions (Fig. 8(d)).

It is unclear that why the scalar QED kink disappears for $d$'s close to 3.[18] One possible explanation is that the numerical convergence becomes harder as $d$ increases, which can be clearly seen by comparing the numerical bounds of $\Lambda = 19$ and $\Lambda = 27$ in Fig. 8(b)-(d). It is also worth noting that, in $d = 3$ dimensions, the numerical bound of $\Delta_s$ is much larger than the value ($\Delta \approx 2$) of the scalar QED. However, based on our numerical data there is no indication that the scalar QED kink will show up in $d = 3$ dimensions as $\Lambda \to \infty$.

---

[17]In $O(2N_f)^*$, $a$ (i.e. $SU(N_f)$ adjoint) corresponds to the rank-2 symmetric traceless tensor of the $O(2N_f)$ WF CFT. Its scaling dimension from $2 + \epsilon$ expansion is $\Delta_a = \frac{2N_f \epsilon}{2N_f - 2} - \frac{2N_f \epsilon^2}{(-2+2N_f)^2} + O(\epsilon^3)$ [71,72].

[18]The scalar QED kink being disappearing shall not be ascribed to the physics of fixed point annihilation as $N_f = 100$ shall be large enough the the scalar QED being conformal in $d = 3$ dimensions.

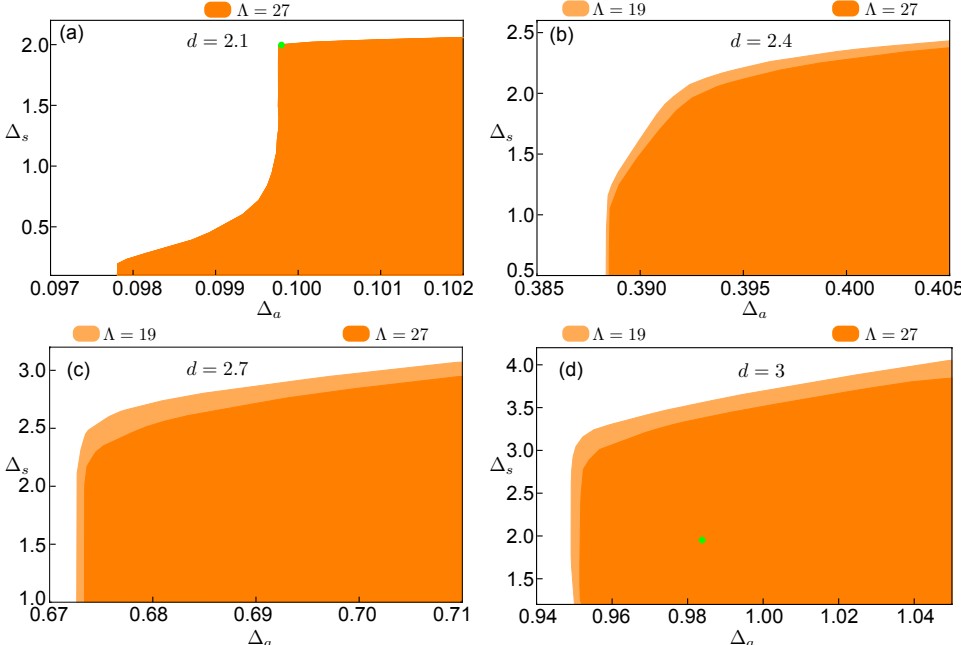

Figure 8: The numerical bounds of $SU(N_f)$ singlet $\Delta_s$ of $SU(100)$ CFTs in $d = 2.1$ (a), $d = 2.4$ (b), $d = 2.7$ (c), and $d = 3$ (d) dimensions. The data is obtained with a gap condition $\Delta_{A\bar{A}} \geq 2(d-2)+1$ imposed, and the allowed regions do not change under a tighter $\Delta_{A\bar{A}}$ gap, e.g. $\Delta_{A\bar{A}} \geq 2(d-2)+1.5$. The green circles mark the scalar QED: (a) it corresponds to the $2 + \epsilon$ expansion results $(\Delta_a, \Delta_s) \approx (0.0998, 1.9998)$; (d) it corresponds to the large-$N_f$ results $(\Delta_a, \Delta_s) \approx (0.984, 1.951)$.

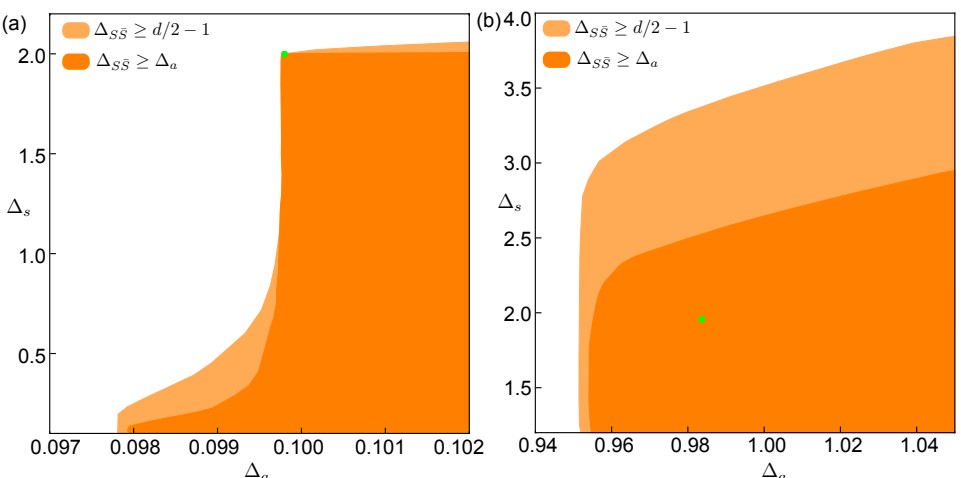

Figure 9: The numerical bounds of $SU(N_f)$ singlet $\Delta_s$ of $SU(100)$ CFTs in $d = 2.1$ (a), and $d = 3$ (b) dimensions. The green circles mark the scalar QED: (a) it corresponds to the $2 + \epsilon$ expansion results $(\Delta_a, \Delta_s) \approx (0.0998, 1.9998)$; (d) it corresponds to the large-$N_f$ results $(\Delta_a, \Delta_s) \approx (0.984, 1.951)$. The light orange and orange feasible regions are obtained with the gap condition i) $\Delta_{A\bar{A}} \geq 2(d-2)+1$, ii) $\Delta_{A\bar{A}} \geq 2(d-2)+1$ and $\Delta_{S\bar{S}} \geq \Delta_a$. The feasible regions do not change under tighter (but still physical) conditions, e.g. $\Delta_{A\bar{A}} \geq 2(d-2)+1.5$ and $\Delta_{S\bar{S}} \geq 1.5\Delta_a$.

A curious observation is that, in $d = 3$ dimensions the numerical bounds are improved significantly by imposing a mild gap $\Delta_{S\bar{S}} \geq \Delta_a$,[19] as shown in Fig. 9(b). In contrast, in $d = 2.1$ dimensions (Fig. 9(a)) by imposing $\Delta_{S\bar{S}} \geq \Delta_a$ the numerical bounds are only improved a little, and the position of the kink does not move. On the other hand, the numerical bounds (for both $d = 2.1$ and $d = 3$) are not further improved under a tighter gap condition, e.g. $\Delta_{S\bar{S}} \geq 1.5\Delta_a$. From the Extremal Functional Method (EFM) [73] we find that on the boundary of feasible region one roughly has $\Delta_{S\bar{S}} \approx 2\Delta_a$, i.e., a relation expected for the scalar QED. Also recall that in Fig. 5, to get the scalar QED island (in the $\Delta_a - \Delta_s$ space) in $2 + \epsilon$ dimensions it is necessary to impose this mysterious gap $\Delta_{S\bar{S}} \geq \Delta_a$. These observations suggest that this gap excludes some crossing symmetric solutions for the bootstrap equations, but we are not able to identify any candidate theory. Nevertheless, in $d = 3$ dimensions with this extra gap imposed the scalar QED kink still does not show up,[20] and the numerical bounds of $\Delta_s$ are still higher than that of the scalar QED. It is possible that one needs to exclude other theories by imposing extra gap conditions in order to spot the scalar QED kink in $d = 3$ dimensions. We leave this for future exploration.

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
