# Peer review of "A roadmap for bootstrapping critical gauge theories: decoupling operators of conformal field theories in $d>2$ dimensions"

_SciPost Physics, doi:SciPost Phys. 11, 111 (2021)_

## Round 2 · Referee Report · Anonymous (Referee 1) · 2021-7-7

Report

The present paper discusses a special class of gauge conformal field theories (CFTs) in general dimensions $d$. Gauge CFTs are defined as IR fixed points of UV gauge theories, which consist of $N_f$ UV scalar or fermion elementary fields transforming in some representation of a gauge group. The local operators in gauge CFTs consist of composite gauge invariant operators made out of those elementary fields. The spectrum of these operators is known for some models in the large $N_f$ or $2+\epsilon$ expansion.

One of the most interesting questions is how to bootstrap gauge CFTs. This is a difficult problem since the information about the gauge group is encoded in the gauge CFT spectrum indirectly (since all the CFT operators are gauge invariant). The authors of this paper provide a recipe for bootstrapping gauge CFTs and explore it in the case of the scalar U(1) QED with $N_f$ elementary fields.

More precisely the idea is to consider a class of all possible gauge CFTs with a similar structure of the spectrum. One can attempt to distinguish a single model out of the class by observing that sometimes some primary operators disappear from the spectrum compared to all the other models (due to two effects: decoupling and equations of motion). Using this information (and some extra reasonable assumptions on the spectrum) in the bootstrap set-up allows to make stronger bounds which either exhibit kinks or even become islands. The obtained kinks and islands in some cases agree very well with the known perturbative $N_f$ or $2+\epsilon$ results.

To the best of my knowledge there is a very limited amount of works dedicated to bootstrapping gauge CFTs. The results of the present paper are certainly very interesting and encouraging for future explorations. The paper is well written. I am thus very happy to recommend it for publication.

Minor linguistic suggestions: 1) p.3: recently, .. becomes $\rightarrow$ recently, .. became 2) In various places (e.g. p.3 and p.19) interested CFTs $\rightarrow$ CFTs of interest.

  • validity: -
  • significance: -
  • originality: -
  • clarity: -
  • formatting: -
  • grammar: -

Author:  Yin-Chen He  on 2021-08-06  [id 1637]

(in reply to Report 1 on 2021-07-07)

We thank the referee for reviewing and recommending our manuscript. We have also followed the referee's linguistic suggestions.

---

## Round 2 · Referee Report · Anonymous (Referee 2) · 2021-8-3

Report

This paper describes a way to bootstrap gauge theories, focusing in particular on the scalar QED theory in three spacetime dimensions. The approach proposed by the authors is to impose well-motivated gaps in the operator spectrum in different channels. These gaps are justified using the equations of motion in a weakly-coupled Lagrangian description. The shortcoming of this approach is that, while the gaps are well-motivated, they cannot be derived in a rigorous way (even in principle), except in weakly-coupled limits such as those provided by the $\epsilon$ expansion and the $1/N_f$ expansion.

The paper focuses on the four-point functions of the lowest-lying operators that transform as an adjoint of the $SU(N_f)$ flavor symmetry. With the gaps imposed, the numerical study that the authors perform yields islands at large $N_f$ and in $2+\epsilon$ dimensions that are consistent with the $1/N_f$ expansion and the $2 + \epsilon$ expansion, respectively.

The paper does a good job explaining the problem, describing the procedure, and presenting the results. I do recommend it for publication with some small changes.

Requested changes

1 - In the Introduction, the sentences "In contrast, gauge theories with distinct gauge groups typically have similar or even identical global symmetries. Their lowest lying operators usually sit in the same representation and have similar scaling dimensions." are too general to be correct. It is not true that any two `typical' gauge theories have a similar spectrum or global symmetries. Perhaps the authors meant that there exist pairs of gauge theories with these properties? I would suggest clarifying this part of the text.

2 - In Eq. (1), $m |\phi|^2$ should be replaced by $m^2 |\phi|^2$.

3 - After Eq. (1), the authors state that the "most fundamental" gauge-invariant operators are boson bilinears. I would say that the operator $F_{\mu\nu}$ should also be part of this list.

4 - The authors say that the theory (1) is also called the $\text{CP}^{N_f - 1}$ model. I think that's not quite precise. The more precise statement is that the theory (1) and the $\text{CP}^{N_f - 1}$ model flow in the infrared to the same CFT fixed point.

5 - In Eq. (9), shouldn't there be a term where the derivative also acts on $\bar \phi$? Same comment for the operators in Table I.

6 - In the paragraph after Eq. (9), the authors state that all operators of scalar QED can be built using $\phi_i$, $\sigma$, and $A_\mu$ and that at large $N_f$ the scaling dimensions add. This is not entirely correct because, in addition to these operators, there are also monopole operators in the CFT.

7 - The scalar QED column of Table I is a bit misleading because there are additional operators that are not listed. For singlet $l = 1$ and $\Delta = 2$, one has the topological current $\epsilon_{\mu\nu\rho} F^{\nu\rho}$ whereas the table lists ``None.'' Similarly for adjoint $\ell=1$ and $\Delta = 3 + O(1/N_f)$, the table again lists ``None'' but one can construct the operator $\epsilon_{\mu\nu\rho} F^{\nu\rho} \bar \phi^i \phi_j - \text{trace}$. I think these operators should be added to the table.

8 - I understand that the gaps imposed distinguish the scalar QED theory from the other thoeries in Table II. However, there are many many other theories in three dimensions with an $SU(N_f)$ flavor symmetry. For example, there could be theories with product gauge groups, bifundamental matter, etc. I am curious whether the authors think that their gap assumptions would exclude all/most other theories beyond those mentioned explicitly in Table II. It would be great if the authors could make some brief comments on this issue.

9 - There are many occurrences of "interested theory". I think the authors mean "theory of interest".

  • validity: -
  • significance: -
  • originality: -
  • clarity: -
  • formatting: -
  • grammar: -

Author:  Yin-Chen He  on 2021-08-07  [id 1643]

(in reply to Report 2 on 2021-08-03)

We thank the referee for reviewing and recommending our paper. We also thank the referee's comments, and below please find our reply.

First of all, we would like to share our viewpoints on the approach we took. It is true that the gaps imposed can only be derived in the weakly-coupled regime, but we think they is a good chance that they are still valid in the strongly-coupled regime. It is because these gaps are physically motivated and capture characteristic features of the gauge group. For example, in the weakly-coupled regime  ($N\gg 1$) of the 2d $SU(N)_k$ WZW CFTs we can analyze decoupling operators semi-classically, and these results are still valid in the  strongly-couple regime (i.e. $N=2$). So we expect  the physics should be qualitatively similar in  $d>2$ dimensions although we cannot prove it rigorously.

Below is our point to point reply, 1.  We changed the sentence to, "In contrast, gauge theories with distinct gauge groups could have similar or even identical global symmetries..." 2.  Corrected. 3.  We have changed it to "One fundamental ...". 4. We have clarified it now. 5. Corrected. 6. We think it is correct to say the monopole operators can be constructed using $A_\mu$, but it is true that their (large-$N_f$) scaling dimensions cannot be obtained by simply adding dimensions of their constituents. We have clarified it now. 7. The operators listed by the referee are parity odd operators (due to the $\epsilon_{\mu\nu\rho}$ tensor), while in this paper we are only concerned with parity even operators for obvious reasons. In the new version of our manuscript  we now made it more explicit that in Table.1 we are classifying parity even operators. 8. Yes, the decoupling operators can be used to exclude these more complicated gauge theories. We have added a footnote (8) to  discuss it. 9. Corrected.

---

## Round 3 · Referee Report · Anonymous (Referee 1) · 2021-8-13

Report

Some minor language corrections have been made. The paper can be published in the present form.

---

## Round 3 · Referee Report · Anonymous (Referee 2) · 2021-10-1

Report

The changes the authors made look good to me, so I recommend this paper for publication.

---

## Editorial Decision

published